# Implementation of HPV Tests in Latin America: What We Learned; What Should We Have Learned, and What Can We Do Better?

**DOI:** 10.3390/cancers14112612

**Published:** 2022-05-25

**Authors:** Luani Rezende Godoy, Júlio César Possati-Resende, Yasmin Medeiros Guimarães, Priscila Grecca Pedrão, Ricardo dos Reis, Adhemar Longatto-Filho

**Affiliations:** 1Molecular Oncology Research Center, Barretos Cancer Hospital, São Paulo 14784-400, Brazil; luanirgodoy@gmail.com (L.R.G.); yasminmguimaraes@outlook.com (Y.M.G.); prigpedrao@gmail.com (P.G.P.); 2Department of Prevention Oncology, Barretos Cancer Hospital, São Paulo 14784-400, Brazil; julio.possati@uol.com.br; 3Department of Gynecologic Oncology, Barretos Cancer Hospital, São Paulo 14784-400, Brazil; drricardoreis@gmail.com; 4Medical Laboratory of Medical Investigation (LIM) 14, Department of Pathology, Medical School, University of São Paulo, São Paulo 01246-903, Brazil; 5Life and Health Sciences Research Institute (ICVS), School of Medicine, University of Minho, 4710-057 Braga, Portugal; 6ICVS/3B’s—PT Government Associate Laboratory, 4710-057 Braga, Portugal; 7ICVS/3B’s—PT Government Associate Laboratory, 4805-017 Guimarães, Portugal

**Keywords:** HPV-based screening, Latin America, cervical cancer, HPV test

## Abstract

**Simple Summary:**

Cervical cancer is caused by HPV and is nearly completely preventable because of the vaccination and screening available. The present review aims to map the initiatives conducted to implement or evaluate the implementation of HPV testing in Latin American countries. We performed a review search on PubMed in the English language and on grey literature in the Spanish language. We found information concerning HPV testing as primary screening in eight countries. We learned that HPV implementation is not only feasible but a very promising tool for reducing cervical cancer morbidity and mortality. The cost for saving lives and reducing suffering due to morbidity must be pragmatically evaluated by the Latin American governments and improving outcomes must become a mandatory priority for those that are responsible for addressing an organized system of screening.

**Abstract:**

Cervical cancer is caused by HPV. Although it is the fourth most common type of cancer diagnosed and the fourth cause of cancer death, cervical cancer is nearly completely preventable because of the vaccination and screening available. The present review aims to map the initiatives conducted to implement or evaluate the implementation of HPV testing in Latin American countries. We performed the review by searching on PubMed in the English language and on grey literature, as most of the information about the guidelines used was found in governmental websites in the Spanish language. We only found information in eight countries concerning HPV testing as primary screening. Only Mexico has established HPV-based screening in all territories. There are three countries with regional implementation. Two countries with pilot studies indicated results that supported implementation. Finally, there are another two countries with a national recommendation. We have learned that HPV implementation is feasible and a very promising tool for reducing cervical cancer morbidity and mortality. The costs associated with saving lives and reducing suffering due to morbidity of a preventable disease must be pragmatically evaluated by the Latin America governments, and improving outcomes must be a mandatory priority for those that are responsible for addressing an organized system of cervical cancer screening.

## 1. Introduction

Cervical cancer (CC) is caused by the persistent infection of oncogenic types of human papillomavirus (HPV), which represents the most common sexually transmitted infection in the world [1]. HPV can be classified as low-risk or high-risk (hr-HPV) according to the oncogenic potential. Currently, the Papillomavirus Family accounts for more than four hundred types; some of them have been identified in different human anatomic sites causing proliferative lesions on the skin and mucosa, or in others such as the penile, anal, oropharyngeal, vulvar, and uterine cervix region [2,3,4]. After an infection in the squamous epithelium of the cervix, HPV can cause precursor lesions named cervical intraepithelial neoplasia (CINs) that are classified into CIN1, CIN2, and CIN3, according to the degree of epithelium injury progression. CIN3 lesions can progress to cancer, particularly if not promptly treated, while mainly CIN1 and CIN2 can regress to normal cytology [5,6,7,8].

Since CC is caused by a virus, it is considered nearly completely preventable [9]. However, among women worldwide, it remains both the fourth most common type of cancer diagnosed and the fourth cause of cancer death. Importantly, these global high rates are the consequences of critical rates even in low and middle-income countries (LMICs), which include Latin American countries, where this type of cancer is the most diagnosed and the leading cause of cancer death among women [9]. According to the World Health Organization (WHO) region division, Latin America comprises 20 countries from North to South: Mexico, Cuba, Guatemala, Honduras, Dominican Republic, El Salvador, Nicaragua, Puerto Rico, Costa Rica, Panama, Bolivarian Republic of Venezuela, Colombia, Ecuador, Peru, Brazil, the Plurinational State of Bolivia, Chile, Paraguay, Argentina, and Uruguay [10]. The reasons the CC is the most diagnosed and the leading cause of cancer death among women in LMICs are related to the social, economic, educational, and geographical barriers that lead to issues in implementing preventable measures [11,12,13,14].

Currently, there are two modalities of CC prevention: primary prevention with HPV vaccines and secondary prevention with screening [9]. There are three HPV vaccines approved by the US Food and Drug Administration (FDA): Gardasil, Gardasil 9, and Cervarix. All vaccines provide protection against HPV 16 and 18. Additionally, Gardasil protects against HPV types 6 and 11 and Gardasil 9 protects against 31, 33, 45, 52, and 58 as well [1]. Regarding screening, there are currently three options available: visual inspection with acetic acid (VIA), cytology (Pap smear), and HPV testing [15,16]. In high-income countries, the decrease in CC rates is due to the availability and effectiveness of preventable measures [17]. Considering the LMICs, due to the issues cited above, the main problems are related to the low coverage of screening programs or the use of non-organized schemes, the low performance of cervical cytology causing low sensitivity, and the difficulty in treatment and follow-up in women with suspect and precursors lesions [11,12,13,14]. In recent past years, a discussion has taken place concerning which method would be better, cytology or HPV testing, considering mortality reduction, early detection of disease, and cost-effectiveness. Regarding the first two aspects, HPV testing demonstrated superiority [18,19]. However, the cost remains impeditive in low-income countries, although initiatives have been made to evaluate the feasibility in these settings.

The present review aims to map the initiatives conducted to introduce or evaluate the implementation of HPV testing in LMICs, specifically in Latin American countries, describing the current status of HPV testing in these localities. We performed the review by searching on PubMed in the English language and also by searching the grey literature, as most of the information about the guidelines used was found on governmental websites in the Spanish language.

## 2. Prevention

In May 2018, the WHO called for a global scale-up of CC control that would lead to the eventual elimination of CC. For this, the aim targets were 90% vaccination among girls up to 15 years, 70% of the population aged 35–45 years screened for HPV, and treatment of 90% of precursor and invasive lesions [20]. Aswe anticipated, there were two ways to prevent: primary prevention that is related to the HPV vaccines that improve the reduction of CC incidence and other HPV-driven cancers, and secondary prevention, which is done with a screening armamentarium looking for precursor lesions, preferentially [9]. The three options for secondary prevention available are visual inspection with acetic acid (VIA), cytology (Pap smear), and HPV testing, depending on the medical infrastructure and medical facilities of the countries where the screening strategies are applied [15,16].

The VIA method is done by the application of 3–5% acetic acid directly to the cervix, and then a visual inspection is performed to identify color modifications, since dysplastic cells dehydrate turning white (acetowhite areas). VIA is more used in LMICs due to the low cost and, if performed properly, can reduce the mortality rate and the incidence of CC. However, this method has a significant number of false-positives and low accuracy in the results with a sensitivity of 84% and specificity of 82% [1,16].

The Doyen method for CC screening, the Pap test (cytology), still represents the gold standard in CC screening for many countries, allegedly due to the low cost and acceptable performance reproducibility, despite its low sensitivity. It consists of a morphological analysis of cervix cells, enabling the identification of dysplastic cells and the grade of this alteration. However, because of the impossibility of implementing Pap tests on a high standard basis for several regions of the globe, new methodologies have been gradually replacing the cytology [1].

HPV testing has been proposed as an alternative screening test due to the higher reproducibility and higher sensitivity. Recent studies have demonstrated that HPV-based screening provides greater protection against invasive CC compared with cytology, and due to its high negative predictive ability, there is the possibility to extend the screening intervals to 5 years [16,21,22,23]. Beyond the extended time of screening, the HPV testing also has the possibility of self-sampling (collecting the material to test for oneself using a vaginal swab or urine samples). Some studies have been demonstrating the effectiveness of the strategy in reaching hard-to-reach women, in addition to the self-sampling, which is associated with higher screening coverage [24,25,26,27,28]. Actually, one of these alternative methods, such as the use of urine to perform the molecular HPV test, has been proposed with very promising preliminary results [27]. Future studies designed to assess the potential value of urine collection as a means of screening for HPC-induced genitourinary lesions may prove the value of this method for use in large populations eligible for HPV screening. Both the longer interval and the self-sampling modality (or urine) are advantages to LMICs, although the higher costs compare with VIA and cytology.

### hr-HPV Test

The American Cancer Society (ACS) recommends HPV testing every 5 years as the first option to screening; another possibility is cotesting (HPV testing in combination with cytology) every 5 years or, if no HPV testing is available, cytology alone every 3 years [29]. The US Preventive Services Task Force (USPSTF), WHO, European Society of Gynecologic Oncology (ESGO), the European Federation of Colposcopy (EFC), and the Brazilian Federation of Gynecology and Obstetrics Associations (FEBRASGO) also recommend HPV testing as a primary screening test rather than VIA or cytology [30,31,32,33]. This recommendation is already implemented or planned to be implemented in several European countries, such as the Netherlands, Turkey, Denmark, the United Kingdom, Belgium [34], and also Australia [35]. In addition, an important problem that has been foreseen is the screening of vaccinated women. In the future, the girls that have been vaccinated will start to have screenings at some point in their lives. Given this, there is a tendency toward a consensus that HPV-based screening will be a better method to screen previously vaccinated women [36].

There are different strategies for HPV-based screening. Some authors have advocated that the more efficient way is using HPV as primary screening, in which the sample is preserved in a liquid medium that can be used for cytology preparation, with the cytology reserved only for women with HPV positive results (known as a reflex-test). Other authors, however, advocate that is preferable to use the same liquid-based cytology sample for both HPV testing and cytology simultaneously. In these cases, if the genotyping is available and the woman is positive for HPV 16 and/or 18, she should be directly referred to colposcopy [37].

The FDA has approved five tests for HPV detection in cytological specimens: the Hybrid Capture 2 HPV DNA test by Qiagen (Hilden, Germany, 2001), Cervista HPV HR test by Hologic (Marlborough, Massachusetts, 2009), Cobas 4800 HPV test by Roche (Basel, Switzerland, 2011), the Aptima HPV assay by Gen-Probe (San Diego, California, 2011, purchased by Hologic in 2012), and the BD Onclarity HPV assay by Becton Dickinson (Franklin Lakes, New Jersey, 2018). Only the Cobas 4800 HPV test and BD Onclarity HPV assay were approved for primary screening. These two tests are automated and based on PCR with the difference in the HPV region analyzed: the first focuses on the L1 region and the second on E6/E7 [38].

The Cobas 4800 HPV test can identify 14 hr-HPV types: a pool with 31, 33, 35, 39, 45, 51, 52, 56, 58, 59, 66, and 68 and genotyping separately of 16 and 18. The sensibility and specificity to CIN2/3 are 92.8–100%, and 24–86.2%, respectively [38].

The BD Onclarity HPV assay can identify 14 hr-HPV types: 16, 18, 31, 33, 35, 39, 45, 51, 52, 56, 58, 59, 66, and 68 simultaneously; discrete individual identification of 16, 18, and 45. The sensibility and specificity to CIN2/3 are 85.7–100%, 17–98.8%, respectively. Additionally, this test has already been evaluated for use in HPV-vaccinated individuals. The sensitivity was lower (80%) and the specificity was higher (52.1%) in vaccinated women than in unvaccinated ones [38].

## 3. Experiences of HPV Testing Implementation in Latin America

CC is a huge health problem in Latin America and the Caribbean (LAC) region. Even with a decrease in the incidence of it in some LAC countries, in general, the incidence and mortality rates remain high, demonstrating that it is necessary to improve the screening programs [39].

The first problem is the coverage of the screening programs and the HPV vaccination. A study demonstrated that only 43.6% of women in LMICs self-reported to have ever been screened for CC, with LAC countries having the highest prevalence (country-level median, 84.6%) [12]. Regarding vaccination, LAC countries demonstrated a median overall coverage for at least one dose is 80%, while it is 55% for the full dose schedule [40,41].

The second one is the lower performance of cervical cytology. In developed countries, the sensitivity is around 50%, while in Latin American countries, sensitivity ranges between 22 and 42% [13,22].

Thirdly, cultural aspects related to pelvic evaluation such as shame, anxiety, the fear of cancer, and poor health literacy affects the impact of cytology-based screening [13,14,42].

Considering these issues, since the year 2000, there has been a discussion concerning t HPV testing as primary screening in Latin America [43] with initiatives in some countries. Below, Figure 1 summarizes the status of HPV-based screening implementation in Latin American countries.

### 3.1. National Implementation

#### Mexico

The first initiative in Mexico was in 2013 when a pilot study was conducted to evaluate the performance and cost-effectiveness of HPV-based screening every five years. The study population was women aged 30–64 years who were tested with hr-HPV genotyping (Cobas 4800 HPV test). Women with a positive result for HPV 16 and/or 18 or abnormal cytology (ASCUS+) were referred to colposcopy [44]. The results showed that HPV 16/18 genotyping with reflex cytology improved the CIN 2+ detection. Furthermore, CIN prevention representsa reduce in cost in comparison with colposcopy services [45].

Then, from 2017 until 2010, another pilot study using the HPV test (hybrid capture 2) as primary screening and using self-collected samples in women aged 25–75 years was conducted. The adjusted positive predictive value of the hr-HPV test was 2.4%, and the negative predictive value was 99.8%. These results suggest that hr-HPV testing in a middle-income country can improve CIN2+ detection [46].

So, currently in México, CC screening is recommended for women between 25 and 64 years, and there are two different modalities according to age: women between 25–34 years are referred to a Pap smear and women between 35 and 64 years are recommended the HPV test. The tests are offered free in all the National Health System centers [47].

### 3.2. Regional Implementation

#### 3.2.1. Argentina

As most of the countries in LAC do, Argentina also recommends the cytology-based screening for women aged 25 and older every three years after two consecutive negative Pap smear tests with a 1-year interval. However, an annual screening is a usual practice. A colposcopy and biopsy are recommended for women with an abnormal cytology (ASCH+) [48,49].

In 2010, Argentina relaunched The National Program on CC Prevention (NPCCP) intending to introduce the HPV-based screening in the Jujuy province. This province had a high CC mortality rate and so it was chosen for the Jujuy Demonstration Project (JDP) [36]. The JDP started with 1 year of planning (from 1 January to 31 December 2011) and 3 years of screening (1 January 2012 to 31 December 2014). On 1 January 2012, all the Jujuy public health institutions changed the primary screening method for CC prevention from cytology-based screening to HPV testing [48].

The HPV-based screening occurred with women over 30 years old. In the screening scheme, the HPV test was collected (Hybrid Capture 2) and a Pap smear, together with the HPV test that was performed first; if the woman was HPV positive, the cytological slide was evaluated. Women with abnormal cytology (ASCUS+) were referred to colposcopy/biopsy; women with normal cytology were guided to re-screening at 18 months. If the HPV test was negative, women were guided to do a re-screening after 5 years [50].

The JDP enrolled 79.196 women; 29.631 women underwent cytology-based screening, and 49.565 women were tested for HPV. Among the women tested for HPV, CIN2+ was diagnosed in 1.4% against 0.8% among women who underwent cytology tests. The JDP also observed other benefits with HPV-based screening, such as the higher adherence to age recommendations (79.3% to the HPV test vs. 98.8% to the cytology test) and a decrease in inadequate samples (3.6% for the HPV test vs. 0.2% for the cytology test). Therefore, the study authors concluded that the HPV-based tests could accelerate the reduction of the CC rate [48]. It is important to notice that, although the HPV tests present good effectiveness, these results were only possible with a high level of support from the political authorities, health authorities, and the stakeholders; as well as the financial support and the program organization efforts [51].

Based on these results, the HPV test screening strategy was definitively implemented in Jujuy and extended to other provinces—Buenos Aires, Neuquén, Catamarca, Tucumán, Misiones, Corrientes, and Chaco—using the same flowchart as the JDP study. The country has 24 provinces, and the strategy offered by the public health system was implemented in 8 (33%) of them. There is the intention for a national implementation over time [49].

#### 3.2.2. Chile

In Chile, cytology-based screening is recommended for women aged 25–64 years to be performed every 3 years [52]. In 2013, Ferreccio et al. conducted a study in public health care centers in Santiago. In it, approximately 8407 women were tested using the HPV test (Hybrid Capture 2) and cytology. The results showed that the HPV test was four times more sensitive for diagnosing lesions and three times more capable of identifying lesions when compared with cytology [53]. As Chile has an organized screening system throughout the country, the authors advocate that the incorporation of HPV-based screening would not be challenging, since they are able to use the infrastructure and support already existing for the Pap smear [54].

In 2015, through the AUGE Clinical Guides Cervical Cancer Uterine (in Spanish *Guías Clínicas AUGE Cáncer Cérvico Uterino*), Chile started to recommend the HPV test as screening, if available, for women aged 30–64 years. For negative results, the test must be repeated in 5 years. The HPV 16 and 18 genotyping is also recommended if possible, and with a positive result, the women are referred for colposcopy [52]. In 2019, the HPV test was implemented in 15 public health care centers, offered at no cost to the patients. The government intended to increase the implementation and in 2020 it would have been in the entire national territory. However, no further information was found regarding the current implementation [55].

#### 3.2.3. Peru

The government of Peru through the Clinical Practice Guideline for the Prevention and Management of Cervical Cancer (in Spanish *Guía de práctica clínica para la prevención y manejo del cáncer de cuello uterino*) in 2017, recommended the HPV test for women aged 30 to 49 years every 5 years if they had negative results. In positive cases, the women are referred to colposcopy or VIA depending on the availability. If the HPV test is not available, the woman is directly referred to VIA. For women between 50 and 64 years old, a Pap smear is recommended every 3 years after two consecutive negative results [56]. Aiming to eliminate CC in Peru by 2030, in the middle of 2021, the Ministry of Health implemented HPV tests in eight regions (Lima, Junín, Loreto, Lambayeque, La Libertad, Cajamarca, Ayacucho, and Arequipa). The intention is progressively to implement the test at the national level [57].

### 3.3. Pilot Test

#### 3.3.1. Brazil

Brazil through the Brazilian Guidelines for the Screening of Cervical Cancer (in Portuguese *Diretrizes Brasileiras para o Rastreamento do Câncer do Colo do Útero*) recommends cytology-based screening for women aged 25 to 64 years every 3 years after two consecutive negative Pap smear tests [58]. If necessary, colposcopy and biopsy are recommended. The estimated number of women in the target age group who undergo Pap testing is lower than that recommended by the international guidelines in some Brazilian regions. In addition, the index of positivity and the positive cytological diagnoses are below the target preconized by the Brazilian National Cancer Institute (INCA) [59].

Barretos Cancer Hospital is located in the interior of the State of São Paulo, and it attends with an oncology service to a vast adjacent region. Established in 2012 in 18 municipalities in Barreto’s region, an organized CC screening is performed using a computerized system, promoting better control of the women’s test status. It enables the institution to send letters inviting all women within the target age who were not tested. Women with repeated abnormal cytology are referred to a colposcopy exam. The HPV test (Cobas Test 4800, Roche) is not used in screening, but since 2014, it has been used to support the follow-up of cases with low-grade lesions (ASCUS, LSIL); if the women are <30 years old, the Pap smear is repeated. If there is a persistent positive result, an hr-HPV test is made; if the women are ≥30 years old, the HPV testing is performed directly. Patients with negative HPV results are recommended to repeat the Pap smear after 12 months [60].

Recently, two initiatives emerged with the intentg to test the HPV-based screening in Brazil under “real-life” conditions [61,62]. The first occurred in São Paulo City (SP) in 2014–2016 with 16.102 women attending the Brazilian public health system. In this study, the hr-HPV DNA test (BD Onclarity HPV assay) and Pap smear were collected simultaneously. The results demonstrated a higher detection of high-grade lesions using HPV-based screening compared with cytology. The authors highlighted the barriers to using this screening modality as the cultural resistance of health professionals and patients’ costumes to the periodic Pap smear as the unique tool for screening. Another issue was the associated cost. However, the implementation of molecular tests should not represent a problem to the Brazilian public health system due to an already established network of laboratories carrying out other viral tests, performing more than 4 million tests per year. Another reason is the high-throughput, fully automated method platforms for HPV testing. So, HPV-based screening shows itself to be feasible and advantageous to being implanted in the Brazilian public health system [61].

The second pilot started in 2017 in Indaiatuba City (São Paulo State) and is still in progress. The authors are following the age recommended by INCA (25–64 years old) and aim to reach 80% coverage in this population. The HPV test (Cobas HPV Test) has been performed as primary screening. If the woman presents a negative result, the test is repeated in 5 years. Different than other initiatives, in this study, the HPV genotype is considered. Women with a positive result for HPV 16 and/or 18 are referred to colposcopy. If the results are positive and other hr-HPV and abnormal cytology is attested, the women are also referred to colposcopy; if the cytology is normal, the HPV test is repeated after 1 year. In 2017, the Mayor of Indaiatuba City approved a law replacing standard cytology screening with HPV test screening in all the city Public Health Care Centers [62]. The preliminary results in 30 months showed that 16.384 HPV tests were performed. The expectation is for the coverage to be increased for more than 80% in the target population at 5 years. Until now, the organized HPV-based screening has demonstrated high coverage and compliance to age recommendations. Beyond the higher detection of CC cases, the majority (67%) have been in the early stage, preceding the diagnosis by 10 years [63]. The cost-effectiveness evaluation showed that the HPV test is low-cost and more cost-effective than the cytology-based screening [64].

All these results support the premise that considers the implementation of HPV tests in primary screening in the Brazilian public health system to be not only feasible but recommendable, although challenging.

#### 3.3.2. Uruguay

Between 2014 and 2017, a pilot study with HPV-based screening was conducted in Uruguay (metropolitan region of Montevideo). A total of 1010 women aged between 30 and 64 years old were enrolled. A unique sample collection was used for the HPV test (hybrid capture 2) and Pap smear. Patients with positive results for the HPV test or the Pap smear (ASC-US+) were referred to colposcopy. The HPV test was more effective as primary screening due to the positivity in 100% of the CIN diagnostic. The authors advocate the necessity to conduct a cost-effectiveness study and increase the population enrolled to verify the feasibility of implementing HPV-based screening [65].

### 3.4. Recommendation

#### 3.4.1. El Salvador

A Pap smear is recommended in the Technical Guidelines for the Prevention and Control of Cervical and Breast Cancer (in Spanish *Lineamientos técnicos para la prevención y control del cáncer cérvico uterino y de mama*) in 2015 for women between 20 and 29 years and aged more than 60 years every 2 years. The HPV test is recommended for those aged 30–59 years every 5 years. If the results are positive, the women are referred to VIA or colposcopy [66]. Although these are the governmental recommendations, we did not find any data concerning the implementation conditions.

#### 3.4.2. Guatemala

A Pap smear is recommended in the Guide for the Prevention, Detection and Treatment of Precursor Lesions of Cervical Uterine Cancer (in Spanish *Guía de atención integral para la prevención, detección y tratamiento de lesiones precursoras de Cáncer Cérvico Uterino*) for women aged 25–54 years every 3 years. The HPV-based screening is recommended for women between 35–45 years every 5 years. If the result of a Pap smear or an HPV test is positive, the women are referred to VIA or colposcopy according to availability [67]. According to the Ministry of Public Health and Social Assistance, between 2015 and 2020,more than 110.000 HPV tests were performed [68].

## 4. ESTAMPA Study

In 2013, the ESTAMPA study recruitment started with the trial registration number: NCT01881659. It is a multicentric study conducted in 12 centers in Argentina, Colombia, Paraguay, Bolivia, Costa Rica, Honduras, Mexico, Peru, and Uruguay. The forecast to finish is July 2022 and the aim is to evaluate the performance of different triage techniques on CC screening. For this, samples to HPV tests and cytology to simulate a reflex-testing are being collected [69].

## 5. Lessons

There are 20 Latin American countries. Of these, only in eight countries was d information about HPV testing as primary screening found. Only Mexico establishes the HPV-based screening in all territories. There are three countries (Argentina, Chile, and Peru) with regional implementation. Two countries with pilot studies (Brazil and Uruguay) indicate results that support implementation. Finally, another two (El Salvador and Guatemala) were found with the national recommendation but without detailed information found about the real status of implementation. It is also possible to observe that the screening interval for HPV tests (5 years) is the same for all countries. However, the target age varies, with the majority starting at 30 years old, as the European and American guidelines recommend. All this information is summarized in Table 1. All the implementations found occur free of cost to the patients. Besides that, all the cost-effectiveness studies showed the feasibility of the implementation of the HPV test as primary screening in LIMCs. The reduction of cost due to the larger screening interval, the smaller number of colposcopy needles, and the wide network present in the public health in some Latin American countries show a good outlooks for implementation. At this point, we have learned that HPV implementation is not only feasible but a very promising tool to reduce CC morbidity and mortality caused by oncogenic HPV persistent infection. The Latin American region is one of the global leaders in a high incidence of CC and a high associated mortality, which identifies the women of these countries as the most important beneficiaries of HPV tests. Improving the efficacy of strategies for this preventable disease is also relevant, because in Latin America, many countries have difficulty in providing the appropriate treatment for advanced-stage CC cases, especially radiotherapy.

## 6. Conclusions—What We Can Do Better

Despite HPV testing showing more effective performance than other screening methods for the reduction of CC mortality and incidence, few countries in Latin America had undergone operational implementation in the field of HPV-based screening programs or, at least, initiated any type of study to evaluate the performance of the HPV test in their communities. We can speculate that this fact demonstrates that the economic, social, and political issues in many of these countries prevent both a well-organized program of continued and robust screening strategy and, concomitantly, give access to the transition from cytology-based screening to more effective methodologies, which could bring advantages such as the increase of the testing interval from 3 to 5 years, enable self-sampling, and improve screening sensitivity. Despite many of these real problems, some countries have succeeded in the implementation of HPV testing, or have conducted studies with good results, and should be seen as an example that improvement in the CC prevention scenario is, although challenging, possible in low-income settings. Considering that CC is a preventable condition and that we have learned that it is possible to improve the screening efficacy using molecular HPV tests even in LMICs, we can do better on three major points: Public Health authorities should be sensitized to consider implementing molecular tests as primary screening; comprehensive initiatives to store complete data should be implemented to guarantee an efficient tracing of women; and, finally, investments in technical education and medical infrastructure should be made to assist women diagnosed with severe lesions. The cost of saving lives and reducing suffering due to the morbidity of a preventable disease must be pragmatically evaluated by Latin America governments and be a mandatory priority for those that are responsible to address the challenge of an organized CC screening system, as many countries have already done on other continents with well-documented success.

## Figures and Tables

**Figure 1 cancers-14-02612-f001:**
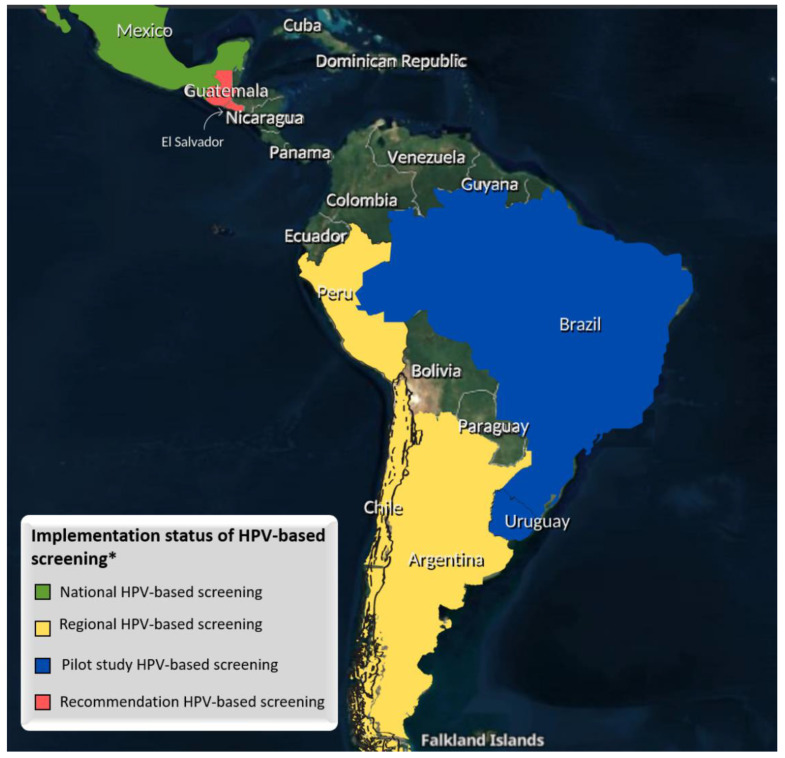
Implementation status of HPV-based screening in Latin American countries. * Countries with information available online.

**Table 1 cancers-14-02612-t001:** Information about HPV-based screening implementation in eight Latin American countries.

Country	Age	Interval	Level	Region	Start Year	Genotyping
México	35–64	5	National			
Argentina	>30	5	Regional	8 provinces	2011	No
Chile	30–64	5	Regional	15 public health care centers	2015	Yes
Peru	30–49	5	Regional	8 regions	2021	No
Brazil	25–64	5	2 Pilot	2 cities	2012/2017	No/Yes
Uruguay	30–64		Pilot	1 city	2014	No
El Salvador	30–59	5	Recommendation		2015	No
Guatemala	35–45	5	Recommendation		2020	No

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
