# Peer review of "Implementation of HPV Tests in Latin America: What We Learned; What Should We Have Learned, and What Can We Do Better?"

_cancers, 2022, doi:10.3390/cancers14112612_

Round 1

Reviewer 1 Report

The Authors map the initiatives in Latin American countries in order to implemented HPV testing as method of screening to prevent the cervix cancer. Only one country (Mexico) screen all territories, 3 countries had regional implementation to other countries had done pilot studies and another 2 had national recomendation. The Authors learned that HPV tests implementation are feasible and very promising to reduce morbidity and mortality due to cervical cancer. However the Authors don´t refer tests in urine as a promising tool to colect samples in low income regions of some of this countries.

Reviewer 2 Report

The manuscript entitled: "Implementation of HPV Tests in Latin America: What we learned; what should we have learned, and what can we do better?" addresses a stringent health issue, that of cervical cancer prevention and screening. It order to improve the Latin-American woman life expectancy and quality of life, and to raise awareness of the authorities in charge, this type of  review papers could be useful.

The present paper, in this form do not fulfill the standards of publication in Cancers MDPI.

Being a review paper, contains too few literature references, even some golden standard papers were omitted, although a solid reference for Latin-American cervical cancer management. The major shortcoming of the present manuscript is that the grey literature is treated as having the same importance as the peer-reviewed scientific literature and analyzed together with the scientific papers. It is uncertain if the authors refer to whitepapers which could provide some data, or simply news published on certain websites. Some of the links in References lead the reader towards a website homepage and not directly to the information, to a deposited pdf, therefore it is extremely difficult to find the content of the indicated sources, for example references 6, 39, 41, 47, 60.   For other resources, authors  provided no electronic address at all, and they are many: 25, 27, 44, 48, 50, 58, 59; it is impossible to reach them. Without a link or more data, the website of journal 43 has an unworkable search engine and without a proper link the reader will fail in finding the pdf of the indicated paper. As well for 55 data are not sufficient to localize the referred article.

Minor comments: the Simple Summary have to be abridged, in the Abstract avoid using repetitively the word "only". The countries where they are national programs for HPV testing should be mentioned as positive example. 

Some phrases have to be reformulated, for example in Introduction chapter: :  "as a necessary cancer cause";  "The reasons for these numbers in LMICs are related to the social, economic, educational (...)" The screening methods should be presented before referring to the consequences of misusing these methods.

Chapter 2: Doyen method with capital D.

Chapter 3: "with studies initiatives in some countries. Bellow, Figure 1 summarizes the status of HPV-based screening implementation in Latin American countries." and similar phrases should be edited.

Chapter 3.1. the first figure in manuscript has in its caption: Figure 2. It is an original figure, or used in the manuscript following an authorization from an Editor?

Chapter 3.2 The exact name of the laws/governmental releases, their exact date could be useful (in addition to the issues mentioned above about the references). Avoid excessive use of past tense.

Chapter 3.3. The official website of the hospitals could be useful, the name/code of the trials o and the publications resulted from these programs.

They are few tables and figures, or schemes in the paper. Table 1 contains a small number of data.

The whole sentence in the Conclusions: "Considering that CC is a preventable condition and that we (...) to assist women diagnosed with severe lesions" have to be reformulated.

Author Response

April 26, 2022

Reviewer 2

The manuscript entitled: "Implementation of HPV Tests in Latin America: What we learned; what should we have learned, and what can we do better?" addresses a stringent health issue, that of cervical cancer prevention and screening. It order to improve the Latin-American woman life expectancy and quality of life, and to raise awareness of the authorities in charge, this type of  review papers could be useful.

The present paper, in this form do not fulfill the standards of publication in Cancers MDPI.

  • Being a review paper, contains too few literature references, even some golden standard papers were omitted, although a solid reference for Latin-American cervical cancer management. The major shortcoming of the present manuscript is that the grey literature is treated as having the same importance as the peer-reviewed scientific literature and analyzed together with the scientific papers. It is uncertain if the authors refer to whitepapers which could provide some data, or simply news published on certain websites. Some of the links in References lead the reader towards a website homepage and not directly to the information, to a deposited pdf, therefore it is extremely difficult to find the content of the indicated sources, for example references 6, 39, 41, 47, 60. For other resources, authors  provided no electronic address at all, and they are many: 25, 27, 44, 48, 50, 58, 59; it is impossible to reach them. Without a link or more data, the website of journal 43 has an unworkable search engine and without a proper link the reader will fail in finding the pdf of the indicated paper. As well for 55 data are not sufficient to localize the referred article.

Answer: Thank you for your suggestion, we added the information about the references. The news or guidelines are governmental information. We cited the scientific literature that we found about the topic, but the majority of this data is in the news or governmental guidelines and this article aims to be the first peer-reviewed to join this data.

Minor comments:

  • the Simple Summary have to be abridged, in the Abstract avoid using repetitively the word "only". The countries where they are national programs for HPV testing should be mentioned as positive example.

Answer: Thank you for your suggestion, we changed the Simple Summary and the Abstract section.

  • Some phrases have to be reformulated, for example in Introduction chapter: : "as a necessary cancer cause";  "The reasons for these numbers in LMICs are related to the social, economic, educational (...)" The screening methods should be presented before referring to the consequences of misusing these methods.

Answer: Thank you for your suggestion, we changed the phrases.

  • Chapter 2: Doyen method with capital D.

Answer: Thank you for your correction, we corrected it.

  • Chapter 3: "with studies initiatives in some countries. Bellow, Figure 1 summarizes the status of HPV-based screening implementation in Latin American countries." and similar phrases should be edited.

Answer: Thank you for your suggestion, we changed the phrase.

  • Chapter 3.1. the first figure in manuscript has in its caption: Figure 2. It is an original figure, or used in the manuscript following an authorization from an Editor?

Answer: Thank you for your correction, we corrected it. It is an original figure.

  • Chapter 3.2 The exact name of the laws/governmental releases, their exact date could be useful (in addition to the issues mentioned above about the references). Avoid excessive use of past tense.

Answer: Thank you for your suggestion, we have included this information.

  • Chapter 3.3. The official website of the hospitals could be useful, the name/code of the trials o and the publications resulted from these programs.

Answer: Thank you for your suggestion, we have included this information.

  • They are few tables and figures, or schemes in the paper. Table 1 contains a small number of data.

Answer: Thank you for your comment, in the table 1 there are the data from the countries that have information available.

  • The whole sentence in the Conclusions: "Considering that CC is a preventable condition and that we (...) to assist women diagnosed with severe lesions" have to be reformulated.

Answer: Thank you for your suggestion, we changed the sentence.

Reviewer 3 Report

The review manuscript (cancers-1670419) entitled “Implementation of HPV Tests in Latin America: What we learned; what should we have learned, and what can we do better?” by Dr. provides a detailed overview on the main initiatives conducted to implement or evaluate the implementation of HPV testing in Latin American countries, including Mexico, Brasil, Argentina, Peru, as well as others. While several improvements can be made in the introductive section, the review manuscript is in general concise, well written and informative and well organized. The work includes important data on the HPV screening programs in Latin American countries.  I therefore recommend a minor revision. I have few suggestions for improving the manuscript:

Major comments
1.    A brief description of the main HPV-driven neoplasms should be included in the introductive section. I would be helpful for the reader as it will increase the importance of the cancer prevention by HPV screening methods such as HPV tests. Besides cercival cancer, several highly aggressive HPV-driven tumors have been identified, such as oropharyngeal, vulvar, penile anal cancer. The authors can check  10.3390/cancers12123772
2.    For completeness of information, the currently employed HPV vaccines, and HPV target types, should be mentioned in the introduction
3.    Regarding the CIN lesions, several premalignant lesions are prone to regress, especially CIN1. This information should be included PMID: 23455757
4.    Acronyms and nouns should be carefully revised throughout the text. For instance, among others, “WHO” has been quoted in line 61 without the corresponding complete name, that is “World Health organization”, while in lines 88 and 105 it is included in both cases as “World Health organization (WHO)”. An additional example can be HR-HPV, that has never been introduced properly as acronym in line 47
5.    Figures, despite being demonstrative, should be improved in the quality

Minor observations
Line 50 regarding mucosal HPVs, I suggest including this important supporting reference (DOI 10.3389/fonc.2019.00355)
Lines 51-55 This sentence is lacking in supporting reference. This recent work detecting several HR-HPVs, mainly HPV16, in CIN lesions, should be included (DOI 10.3389/fmicb.2020.591452)
Linbe 61 WHO should be World Health organization (WHO) when mentioned for the first time in the text. Accordingly, it should only be WHO in lines 88 and 105
Line 93 “reduction of CC incidence”. I would include “and other HPV-driven cancers”
Line 162 an additional consequence of the high frequency of cervical cancer in latin America is the low HPV-vaccine coverage PMID: 30699273 PMID: 31099692
Line 313 better “A total of 1010”
